# The Influence of Skin Thickness on Flash Glucose Monitoring System Accuracy in Dogs with Diabetes Mellitus

**DOI:** 10.3390/ani11020408

**Published:** 2021-02-05

**Authors:** Francesca Del Baldo, Alessia Diana, Claudia Canton, Nikolina Linta, Roberto Chiocchetti, Federico Fracassi

**Affiliations:** Department of Veterinary Medical Sciences, University of Bologna, Ozzano dell’Emilia, 40064 Bologna, Italy; francesca.delbaldo2@unibo.it (F.D.B.); claudia.canton@studio.unibo.it (C.C.); nikolina.linta2@unibo.it (N.L.); roberto.chiocchetti@unibo.it (R.C.); federico.fracassi@unibo.it (F.F.)

**Keywords:** canine diabetes mellitus, continuous glucose monitoring system, flash glucose monitoring system, skin thickness, ultrasound

## Abstract

**Simple Summary:**

A flash glucose monitoring system (FGMS) has been validated for use in dogs with diabetes mellitus and diabetic ketoacidosis. It continuously measures the glucose in the interstitial fluid through a small filament (5 mm long) inserted under the skin. Interstitial glucose concentrations are reportedly comparable to whole blood glucose concentrations. However, several factors can influence the performance of interstitial sensors, including the proportion of interstitial fluid in a tissue. The influence of skin thickness on flash glucose monitoring system accuracy has not been investigated in previous studies; therefore, the aim of this study was to evaluate whether FGMS accuracy is affected by skin thickness. On the basis of our results, skin thickness seems to affect FGMS measurements; the mean bias was significantly inversely correlated (*p* = 0.02; *r* = −0.6) with the mean skin thickness, and clinical accuracy according to ISO 15197:2013 criteria was observed only in dogs with skin thickness > 5 mm, with 99% of the results falling in zone A + B of the Parkes consensus error grid analysis. In dogs with thin skin (<5 mm), the clinical accuracy was low, and the results should be interpreted with caution.

**Abstract:**

A flash glucose monitoring system (FGMS) has been validated for use in diabetic dogs. However, it is unknown whether skin thickness affects FGMS measurements. The aim of this study was to evaluate whether FGMS accuracy is affected by skin thickness. Fourteen client-owned diabetic dogs on insulin treatment were prospectively enrolled in the study. The dogs were divided into two groups according to their ultrasound-measured skin thickness: dogs with skin thickness < 5 mm (Group 1) and dogs with skin thickness > 5 mm (Group 2). On days 1, 7 and 14, glucose curves were obtained simultaneously using the FGMS and a validated portable blood glucose meter. Paired measurements were used to calculate the mean bias and to determine accuracy according to ISO 15197:2013 criteria. The mean bias was significantly inversely correlated (*p* = 0.02; *r* = −0.6) with the mean skin thickness. Clinical accuracy was observed only in Group 2, with 99% of the results in zone A + B of the Parkes consensus error grid analysis. In conclusion, skin thickness seems to affect FGMS measurements, and the device is accurate in dogs with thicker skin (>5 mm); in dogs with thin skin (<5 mm), the clinical accuracy is low, and the results should be interpreted with caution.

## 1. Introduction

The flash glucose monitoring system (FGMS) is a novel continuous glucose monitoring system (CGMS) recently validated for use in dogs with diabetes mellitus and diabetic ketoacidosis [1,2]. It consists of a 35 mm × 5 mm round sensor, which continuously measures the glucose in the interstitial fluid through a small filament inserted under the skin, allowing more rapid glucose monitoring. The system has several features that distinguish it from existing sensor technology. It provides comprehensive glucose data without the need for calibration, and it can be worn for up to 14 days. The hand-held reader is used to scan the sensor to receive a glucose result along with historic results with a 15-min frequency for up to 8 h.

Interstitial glucose (IG) concentrations are reportedly comparable to whole blood glucose concentrations in humans and other animals as a result of an equilibrium that develops between the glucose concentration in the blood and in the interstitium [3,4,5,6,7,8,9,10,11]. However, several factors can influence the performance of interstitial sensors, including the proportion of interstitial fluid in a tissue, which is lower and more variable in subcutaneous tissue and dependent on the size of adipocytes and subsequent adipocyte blood flow [12]. Moreover, in humans, tissue glucose concentration nadirs in muscle have been reported to be delayed in time and lower in magnitude relative to glucose concentrations in adipose tissue and blood, especially during insulin-induced hypoglycemia [13,14]. Therefore, decreased thickness of the subcutaneous adipose tissue layer may result in closer sensor proximity to the underlying muscle tissue and, consequently, in inaccurate glucose concentration results. Despite the evidence that the FGMS is accurate for evaluating IG in dogs, the influence of skin thickness on sensor readings has not been reported in the veterinary literature. Therefore, the aim of this study was to assess the influence of skin thickness on FGMS measurements.

## 2. Materials and Methods

### 2.1. Diabetic Dogs

Fourteen client-owned diabetic dogs on insulin treatment were included in the study. The median age was 10 years (7–14 years). The median body weight was 9.9 kg (5.3–30.6 kg), and the median body condition score (BCS) was 5 (4–8). The median hematocrit was 44% (39–55%). There were 8 mixed-breeds, 2 English Setters and one each of the following breeds: Pinscher, Bichon Poodle, Maltese and Labrador Retriever. Three dogs had concurrent hypothyroidism and were receiving levothyroxine (Canitroid, Dechra Pharmaceuticals, Northwich, UK) supplementation, while 4 dogs had concurrent pituitary-dependent hypercortisolism and were receiving trilostane (Vetoryl, Dechra Pharmaceuticals, Nortwich, UK). Table 1 shows the most relevant clinical data of each dog.

Based on skin thickness measured by ultrasonography (US) at T0 and according to the length of the sensor filament (5 mm), the dogs were divided into two groups: dogs with a skin thickness < 5 mm (Group 1) and dogs with a skin thickness > 5 mm (Group 2).

The study protocol was approved by the Ethical Committee of the University of Bologna (ID1009/18), and informed consent was obtained from the owners.

### 2.2. Data Collection

Three separate glucose curves (GCs) were obtained for each dog during the recording period of their respective FGMS. On days 1, 7 and 14, paired in-hospital GCs were obtained using the FGMS (FreeStyle Libre, Abbott, UK) and a validated portable blood glucose meter (PBGM) (Optium Xceed, Abbott, UK) device [15]. This PBGM underestimates the real BG concentration in the euglycemic and hyperglycemic range, with a mean difference between capillary BG measured by the PBGM and serum BG obtained using the reference method (hexokinase method) of −0.26 mmol/L (4.8 mg/dL) and −4.11 mmol/L (74 mg/dL), respectively. Conversely, in the hypoglycemic range, this PBGM overestimates the BG concentration, with a mean difference of 0.37 mmol/L (6.8 mg/dL) [15]. On day 1 of the study, the dogs were hospitalized after food and insulin were given at home. Immediately after arrival in the hospital, an ultrasound (US) of the dorsal side of the neck (*regio colli dorsalis*) was performed, and immediately after that, the sensor was placed in the same clipped and cleaned area. In particular, an area of around 5 × 5 cm, halfway between the margin of the occipital bone and the dorsal margin of the scapula, was clipped and cleaned with chlorhexidine and 70% alcohol. The sensor was inserted with the aid of the applicator provided by the manufacturer. After positioning, it was fixed with extra tape (PIC Solution Soft Fix, Pikdare S.p.A., Casnate con Bernate, Como, Italy), and a cotton and an elastic body bandage (Vetrap RM, 3M Italia Srl, Pioltello, Milano, Italy) were used to secure the sensor to the body [1]. The sensor measures the IG concentration through a small catheter (5 mm long × 0.4 mm wide) inserted under the skin. The sensor has a 1-h period of initialization. The detection limits range from 1.1 to 27.7 mmol/L (20 and 500 mg/dL); when the IG concentration is 1.1 mmol/L (≤20 mg/dL) and ≥27.7 mmol/L (500 mg/dL), the reader shows “LO” (≤1.1 mmol/L; 20 mg/dL) and “HI” (≥27.7 mmol/L; 500 mg/dL), respectively. For a total of 10–12 h, IG measurements were recorded using the FGMS on a two-hourly basis. Capillary blood glucose was obtained from the pinna every 2 h using the PBGM during the same period. On days 7 and 14, food and insulin were given at home; the paired GCs were started after the dog arrived at the clinic (≤1 h after insulin administration) using the same protocol. The paired measurements thus obtained were used to calculate the mean bias and to determine the analytical and clinical accuracy of the FGMS in the two groups of dogs using the ISO 15197:2013 criteria.

### 2.3. Ultrasonography of the Neck

Ultrasonography of the dorsal skin of the neck was carried out by the same experienced sonographer (AD) using a real-time ultrasound machine (Epiq 5G Ultrasound System; Philips Healthcare, Monza, Italy) equipped with a broadband high-frequency linear-array transducer (5–18 MHz) immediately before the sensor application (T0) and after its removal (T1) in all dogs. The transducer was placed in the area previously described in the data collection section. The gain control was set at 36–40%, and the focal zone was placed at the level of the skin. Ultrasound examination of the skin was carried out as previously described [16,17]. Longitudinal and transverse scans were obtained, and the images of the skin were recorded in DICOM format for subsequent offline analysis. The description and the measurements of the US images were obtained using specific imaging software (OsiriX Imaging Software; Pixmeo, Geneva, Switzerland). The appearance of the US skin pattern of each dog was compared to the normal aspect of the canine skin pattern previously reported [16,17]. Measurements of the skin were made starting from the outer side of the skin surface (epidermal echo entry-E) to the clearly recognizable acoustic interface between the dermal (D) and the subcutaneous (S) layers (Figure 1). Considering the absence of statistical difference between measurements obtained in longitudinal and transverse planes, the mean value of the 3 measurements obtained in both the longitudinal and the transverse planes for the two different time points (T0 and T1) was used for the statistical analysis.

### 2.4. Statistical Methods

The data were analyzed using commercial statistical software packages (GraphPad Prism 7^®^, San Diego, California, USA and R software version 3.6.2, R core Team, Vienna, Austria). Data distribution was evaluated using the D’Agostino and Pearson tests, and parametric or non-parametric tests were used accordingly. Non-normal data are reported as medians and ranges, while normal data are expressed as the mean ± standard deviation (SD). Skin thickness values at T0 were correlated to the BCS using the Spearman test. Skin thickness values at T0 and T1 were compared using the paired *t*-test in cases where both measurements were available. Moreover, the Mann–Whitney *U* test was used to compare the skin thickness between Group 1 and Group 2. The bias, defined as the absolute value of the difference between interstitial glucose concentration obtained with the FGMS and corresponding blood glucose concentrations obtained with the PBGM, was calculated. For each dog, the mean bias was calculated as the mean of the biases obtained from all the paired values, and it was correlated to the mean skin thickness using the Pearson test. The unpaired *t*-test was used to compare the mean bias between Group 1 and Group 2. The Mann–Whitney *U* test was used to compare the body weights between Group 1 and Group 2. *p* < 0.05 was considered significant.

Analytical and clinical accuracy within the 2 groups of dogs was evaluated by comparing the results of the PBGM measurements and those obtained from the FGMS using the ISO 15197:2013 criteria. Both of the following minimum criteria for acceptable system accuracy should be met: (1) 95% of the results must be within ±0.8 mmol/L (15 mg/dL) of the BG concentration for a BG concentration < 5.5 mmol/L (100 mg/dL) and within ±15% of the BG concentration for a BG concentration > 5.5 mmol/L (100 mg/dL) and (2) 99% of the individual BG measured values should fall within zones A and B of the Parkes consensus error grid analysis (EGA) for type 1 diabetes mellitus (DM) [18]. Fisher’s exact test was used to compare the percentage of IG values fulfilling the ISO 15197:2013 requirements between Group 1 and Group 2. Significance was set at a value of *p* < 0.05.

## 3. Results

All sensors reported IG concentrations within 60 min post-application. In 5/14 dogs, the sensor recorded for 14 days, while in 9/14 dogs, the sensor stopped recording IG before 14 days due to accidental detachment (2/14) or because the hand-held reader persistently showed “LO” or “ERR” (the reader is unable to communicate with the sensor) (7/14). In these dogs, the recording period of the sensor was 13 days in 2/14, 12 days in 1/14, 10 days in 4/14, 6 days in 1/14 and 2 days in 1/14. Ultrasonographic examination of the skin was available in 14/14 and 11/14 dogs at T0 and T1, respectively (Table 2).

In 3/14 dogs, US examination after sensor removal was not performed due to poor owner compliance. In 8/11 dogs, the US examination was carried out immediately after sensor removal on day 14. Among these, there were the 5 dogs in which the sensor lasted for 14 days, the 2 dogs in which the sensor lasted for 13 days and the 1 dog in which the sensor lasted for 12 days. In 2/11 dogs, the sensor was detached and therefore removed by the owner on day 10, and the US evaluation was carried out 4 days later. In one dog, the sensor was removed on day 6, and the US was performed on day 7. At T1, 3/14 dogs showed mild self-limiting erythema at the point of the sensor application (Appendix A).

The ultrasonographic features of the canine skin showed a consistent pattern characterized by three distinct layers (i.e., epidermal entry echo, dermal layer and subcutaneous layer) in all dogs and at both time points. At T0, an irregular entry echo and double-layered appearance of the dermis were appreciable in 5/14 dogs and in 7/14 dogs, respectively (Figure 1) (Table 2).

At T1, the epidermal entry echo was irregular in 7/11 dogs, and 5/11 dogs showed the double-layered appearance of the dermal layer. The skin thickness was not correlated with the BCS (*r* = 0.48; *p* = 0.08). Mean skin thickness was 5 ± 1.4 and 5 ± 1.8 mm at T0 and T1, respectively. The difference was not significant (*p* = 0.84). Six dogs had a skin thickness < 5 mm (Group 1), while eight dogs had a skin thickness > 5 mm (Group 2). The median skin thickness was 3.3 mm (3.0–4.9) in Group 1 and 5.9 mm (5.5–6.9) in Group 2. The difference was significant (*p* = 0.0007). Group 1 included the following breeds: 1 Pinscher, 1 English Setter, 3 small mixed-breeds and 1 medium mixed-breed. Group 2 included 1 English Setter, 1 Labrador Retriever, 1 Maltese, 1 Bichon Poodle, 2 small mixed-breeds and 2 large mixed-breeds. The median weight in Group 1 was 6.05 kg (5.4–16.7), and in Group 2, it was 17.05 kg (6.5–30.6) (*p* = 0.04).

A total of 127 paired glucose measurements were available for analysis, of which 48 were obtained from Group 1 and 79 were from Group 2. Considering all the dogs, the mean bias between the glucose concentrations obtained using the FGMS versus those obtained using the PBGM was 3.4 mmol/L (±2.11) (61 mg/dL ± 38), and the biases were significantly inversely correlated with the skin thickness (*r* = −0.6; *p* = 0.02) (Figure 2).

The mean bias of the FGMS versus PBGM in Group 1 was 4.1 (±2.6) (74 mg/dL ± 47), and in Group 2, it was 0.33 mmol/L (±3.77) (6 mg/dL ± 68). The difference was significant (*p* < 0.0001).

Considering the ISO 15197:2013 requirements, in Group 1, the percentages of values within ±0.83 mmol/L (15 mg/dL) of the BG concentration for BG concentration < 5.55 mmol/L (100 mg/dL) and within ±15% of the BG concentration for BG concentration ≥ 5.5 mmol/L (100 mg/dL) were 0% (0/4) and 32% (14/44), respectively (Figure 3A). In Group 2, the percentages of values within ±0.83 mmol/L (15 mg/dL) of the BG concentration for BG concentration < 5.5 mmol/L (100 mg/dL) and within ±15% of the BG concentration for BG concentration > 5.5 mmol/L (100 mg/dL) were 67% (4/6) and 48% (35/73), respectively (Figure 3B). These percentages were significantly different (*p* = 0.03).

Evaluation of the data using the Parkes consensus EGA showed that 94% (45/48) and 99% (78/79) of the FGMS results fell in zones A + B in Groups 1 and 2, respectively (Figure 4A,B). These percentages did not differ significantly (*p* = 0.15).

## 4. Discussion

The results of this study reveal that skin thickness can affect FGMS readings and, in particular, that the accuracy of the device is higher in dogs with thicker skin.

All the dogs in this study showed the typical US skin pattern characterized by three echogenic layers as previously described in clinically normal dogs [16,17,19]. The irregularity of the epidermal entry echo detected in several dogs at both time points might reflect skin irritation induced by the clipper, corneal desquamation or inflammation induced by the sensor. In several dogs, a two-layered appearance of the dermis was evident. In humans, this aspect reflects a demarcation between the papillary dermis (containing fine connective tissue) and the reticular dermis (composed of large bundles of fibers in a water-rich interstitial medium) [20]. Dermal echogenicity depends on the amount of collagen fibers (which increase dermis echogenicity) and water influx (which decreases dermal echogenicity), probably through distension of the fiber network [21,22]. Therefore, the two ultrasonographic patterns identified in the dermal layer of the dogs reported herein might be related to the differing amounts of dermal fluid storage (different hydration state), which influences its echogenicity [17]. Unfortunately, the hydration state of the dogs was not assessed in this study, preventing this evaluation.

Mean skin thickness during the wearing period ranged from 5 ± 1.4 mm (T0) to 5 ± 1.8 mm (T1); differences between the two time points were not observed. This evaluation was carried out in order to detect whether inflammation induced by the FGMS sensor could influence skin thickness and, consequently, FGMS accuracy during the wearing period of the sensor. It can be assumed that, even if a mild inflammation develops, skin thickness is not affected. Skin thickness in dogs is reported to vary from 0.5 to 5.0 mm depending on several factors, such as the cutaneous site [23] and the state of hydration [17], as well as individual factors related to breed, age and sex [23,24,25]. The average skin thickness in this study was greater than has previously been reported [16,17,26,27,28,29]. A possible explanation is that the dorsum of the neck is one of the areas reported to have the thickest skin [23,24]. Further, previous studies in which skin thickness in dogs was investigated were performed in healthy dogs or in dogs with hyperadrenocorticism, an endocrinopathy associated with a reduced skin thickness [29]. In contrast, our study includes dogs with naturally occurring DM, which, in humans, is associated with *scleredema diabeticorum* [30], a dermatologic complication characterized by thickening of the skin affecting the posterior neck. Although this dermatologic complication has never been documented in diabetic dogs, similar skin changes might affect diabetic dogs too and, therefore, might have been the reason for the higher skin thickness reported in this population of diabetic dogs.

Considering that the length of the sensor is 5 mm, our results suggest that the sensor is also capable of reading IG at the dermal level instead of the subcutaneous level, as described for human beings, where the skin thickness at the level of sensor application (lateral aspect of the arm) is approximately 1 mm [31]. However, this is only speculation because the location of the sensor in the dermis was not confirmed. Indeed, due to technical reasons (i.e., reverberation artifact created by the metallic filament and limited skin surface), it was not possible to perform a US after the application of the sensor. The median body weight in Group 2 was significantly higher as compared to Group 1. However, both groups included small- and large-breed dogs, and BCS was not correlated with the mean skin thickness. Hence, it can be assumed that there is a wide individual variability in the skin thickness, which is difficult to predict on the basis of the breed and size or BCS of the dog.

An inverse correlation between mean bias and mean skin thickness was found; moreover, the mean bias in Group 1 was significantly higher as compared to Group 2. In both Group 1 and Group 2, analytical accuracy based on ISO 15197:2013 requirements was not obtained. However, in Group 2, a significantly higher percentage of IG values fell within the desired ranges. A previous study found better results for the analytical accuracy of the FGMS in stable diabetic dogs [1], while the results in the present study are similar to those seen in another study in which the FGMS was used in unstable diabetic dogs [2]. These differences may be due to the fact that, in the study of Corradini et al., the accuracy of the FGMS was evaluated by comparing the glucose measurements of the FGMS with the plasma glucose measured using the hexokinase method, whereas, in the present study, as well as in the Malerba et al. study, the same PBGM was used as a reference. Human PBGMs, like the one used in this study, are plasma calibrated to enable a better comparison between laboratory and PBGM device measurements. As the amount of glucose within erythrocytes in dogs is lower than that in humans, the human PBGM device tends to underestimate the “true” BG concentration [32]. Therefore, in this study, the use of PBGM as reference method might have resulted in a greater overestimation and lower underestimation of the IG values.

The Parkes EGA showed good clinical accuracy for Group 2, with 99% of the FGMS readings in zones A + B, while the clinical accuracy was lower in Group 1, with 94% of FGMS readings in zones A + B. Considering all of the above results, it can be assumed that the accuracy of the FGMS is lower in dogs with thinner skin, although ISO 15197:2013 requirements were not completely fulfilled in either group of dogs. A possible explanation could be the fact that, in dogs with a greater skin thickness, the sensor measures the IG in the reticular dermis (instead of the interstitial subcutaneous fluid), which is rich in water as well as in arteriolar and venular supply [22], and it contains a third of the interstitial fluid volume [33]. According to human studies, the hypodermis is not an ideal location for reliable and fast glycemia tracking due to its heterogeneity and the scarce local distribution of capillaries compared to the dermal region [34]. Indeed, interstitial fluid is more abundant in the dermis, and dermal glucose concentration dynamics more closely follow oscillation in the blood [35,36]. Hence, in dogs with thicker skin (>5 mm), the sensor reaches the dermis (instead of the subcutaneous tissue), where there might be a more rapid equilibrium due to the presence of the microcirculatory bed [23] between the blood glucose and the IG, which makes the FGMS readings more accurate. In humans, no studies have evaluated the influence of skin thickness on FGMS accuracy, and FGMS readings did not appear to be affected by the BMI [37]. However, the accuracy of a different CGMS was slightly greater in subjects with a higher BMI than in those with a lower BMI [38]. A possible explanation is that, in human patients with lower BMI values, the reduced thickness of the subcutaneous adipose tissue layer may result in closer sensor proximity to the underlying muscle tissue [38]. Tissue glucose nadirs in muscle have been reported to be delayed in time and reduced in magnitude relative to glucose in adipose tissue and blood, especially during insulin-induced hypoglycemia [13,14]. The differences in adipose tissue blood flow observed in subjects with different BMIs may also contribute to the apparent effect of the BMI on accuracy [38,39]. Nevertheless, these considerations are likely not applicable to the findings in the present study since, as stated above, in dogs, the sensor should reach the dermal layer instead of the subcutaneous tissue where the adipose tissue is located.

At T1, a moderate erythema at the point of the sensor application was observed in three dogs. In all dogs, the skin reaction was self-limiting without the need for any medication. In dogs, the development of a mild erythema at the site of the sensor application has been reported in 50% of dogs [1] and in 7% of dogs [2]. In humans, skin reactions to the FGMS have been described in several reports [40,41,42,43,44] and can be irritative or allergic skin lesions. Irritation is more common, and it is usually due to individual and physical factors [45]. Allergic contact dermatitis is less common but of greater clinical significance [46]. It is thought to be due to the isobornyl acrylate contained in the sensor itself, which can migrate into the adhesive part of the device and as such comes into contact with the skin, leading to an allergic type 4 reaction [42].

There were several limitations in the present study. First, capillary BG concentrations obtained using a PBGM not calibrated for dogs, and not using the hexokinase method, were used as a reference to evaluate the accuracy of the FGMS. The PBGM used is known to underestimate the real BG concentration and to be affected by the low hematocrit (anemic dogs had greater readings than BG concentration measured by the reference method) [15]; however, in this study, no dog was anemic. Second, the small number of dogs included in the study could have influenced the power of the statistical analysis and might have affected the reliability of the results. According to ISO 15197:2013, evaluation of system accuracy should be performed with at least 100 capillary blood samples from different subjects, and the glucose levels in these samples should be distributed into defined glucose concentration intervals between ≤2.77 mmol/L (50 mg/dL) and >22.2 mmol/L (400 mg/dL). In this study, no samples in the hypoglycemic range were included, and the number of samples for each group was less than 100. However, the accuracy of the FGMS in dogs has already been studied [1]. Third, the authors did not evaluate whether the accuracy of the system varied during the entire wearing period. However, some studies from human medicine have demonstrated that the accuracy of the device remains stable during the 14 days of use [37,47,48,49], although another study showed that after 1 week of use, the glucose sensors produce an inflammatory reaction, which seems to compromise their accuracy [50]. Furthermore, in the previous study in which the FGMS was used in diabetic dogs, the mean absolute relative difference (MARD) on day 14 was only slightly higher as compared to the MARD on day 1 [1]. Considering that, in this study, skin thickness did not change during the wearing period of the sensor, the development of mild skin inflammation should not affect FGMS accuracy. Further, the hydration state can affect skin thickness and FGMS accuracy [51]; however, in this study, the hydration state of the dogs was not assessed, preventing additional analysis. Finally, concurrent diseases such as hypercortisolism and hypothyroidism could interfere with glycemic control, contributing to glycemic variability and influencing FGMS accuracy regardless of skin thickness. However, both groups included dogs with concurrent diseases (four dogs in Group 1 and three dogs in Group 2), so concurrent diseases are unlikely to be responsible for the decreased accuracy of the FGMS detected in dogs with skin thickness < 5 mm.

## 5. Conclusions

In conclusion, the FGMS readings seem to be affected by skin thickness and, in particular, dogs with thicker skin have more accurate readings as compared to dogs with thinner skin. Considering the difficulty in predicting skin thickness on the basis of dog size and BCS, the authors recommend checking BG concentration whenever unexpected FGMS results are obtained.

## Figures and Tables

**Figure 1 animals-11-00408-f001:**
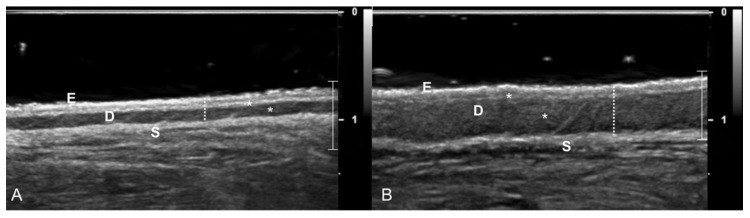
Ultrasonographic appearance of the neck skin of a dog with a skin thickness < 5 mm (**A**) and a dog with a skin thickness > 5 mm (**B**). Three distinct layers are clearly recognizable: the epidermal echo entry (E), the dermis (D) and the subcutaneous tissue (S). Note that the echogenicity of the second layer (D) is not uniform, and two distinct bands (*) with different echogenicities are recognizable. Measurements of the skin (the dotted lines) showed notably different skin thicknesses between the two dogs. The focal zone (white bar) and the graduated scale (0–1 = 1 cm) are displayed on the right of each image.

**Figure 2 animals-11-00408-f002:**
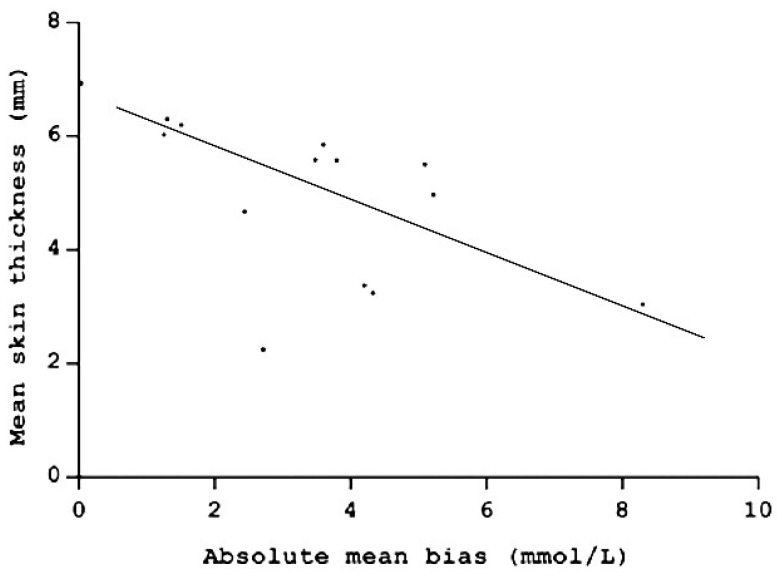
Correlation between skin thickness and absolute mean bias between the glucose concentrations obtained using the flash glucose monitoring system (FGMS) versus those obtained using the portable blood glucose meter (PBGM) of each dog.

**Figure 3 animals-11-00408-f003:**
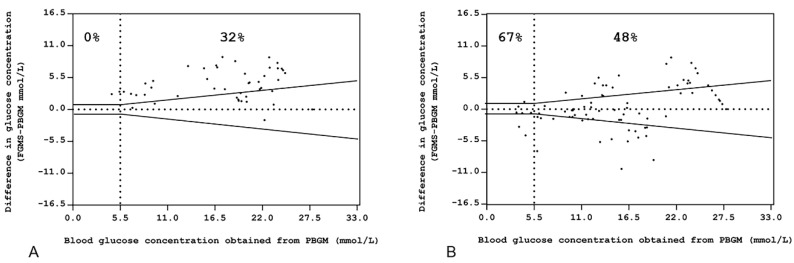
Bland–Altman plots represent the differences between the glucose concentrations obtained by the use of the FGMS versus those obtained using the PBGM in (**A**) Group 1 and (**B**) Group 2. The PBGM glucose values plotted against absolute errors for each corresponding value are on the *x*-axis. The standard required limits are defined by the black symmetric line: at ±0.83 mmol/L (15 mg/dL) from the reference value for glucose determinations < 5.5 mmol/L (100 mg/dL) and ±15% from the reference value for glucose determination > 5.5 mmol/L (100 mg/dL). Percentages express the number of samples within the limits when the reference determination was < or >5.5 mmol/L (100 mg/dL) and for the total number of measurements (central % value).

**Figure 4 animals-11-00408-f004:**
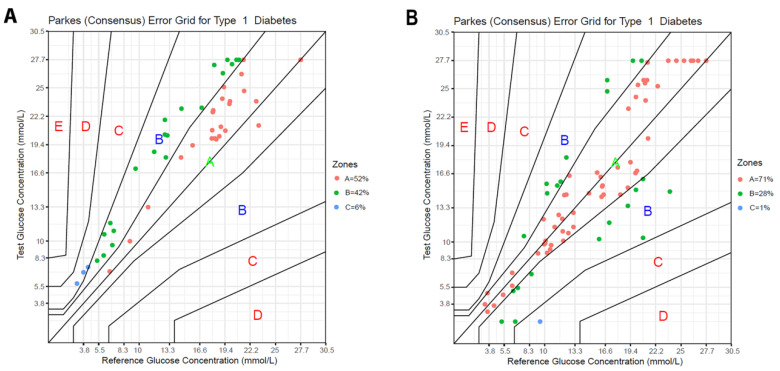
Parkes consensus error grid analysis (EGA) representation of the percentage of values within the different zones in (**A**) Group 1 and (**B**) Group 2 dogs. The reference glucose values (blood glucose obtained by a PBGM), on the *x*-axis, are plotted against the interstitial glucose (IG) measurements obtained by the FGMS, on the *y*-axis. The different zones designate the magnitude of risk: no effect on clinical action (zone A); altered clinical action—little or no effect on the clinical outcome (zone B); altered clinical action—likely to affect the clinical outcome (zone C); altered clinical action—could have a significant medical risk (zone D); and altered clinical action—could have dangerous consequences (zone E). ISO 15197:2013 requires that 99% of the values fall within zones A + B for a device to be considered accurate.

**Table 1 animals-11-00408-t001:** Clinical data, skin thickness and absolute mean bias of each dog.

	Age	Breed	Weight (kg)	BCS	Concurrent Disease	Initial Skin Thickness (mm)	Final Skin Thickness (mm)	Absolute Mean Bias (mmol/L)
**Patient 1**	9y5m	Pinscher	6	5	/	3.045	3.073	8.33
**Patient 2**	12y2m	English Setter	16.7	5	Hypothyroidism	3.376	3.328	4.21
**Patient 3**	9y5m	Small mixed breed	11.8	5	/	6.303	5.286	1.32
**Patient 4**	10y5m	Bichon Poodle	7.4	6	/	5.583	5.255	3.79
**Patient 5**	7y2m	Maltese	6.5	6	Hypercortisolism	5.510	4.950	5.11
**Patient 6**	7y8m	Large mixed breed	26.5	8	Hypothyroidism	5.858	7.770	3.60
**Patient 7**	11y	Large mixed breed	30.6	6	/	6.028	/	1.25
**Patient 8**	13y8m	Small mixed breed	6.1	5	/	4.675	5.306	2.44
**Patient 9**	13y2m	English setter	22.3	6	/	6.201	7.023	1.51
**Patient 10**	9y8m	Small mixed breed	5.3	5	Hypercortisolism	4.978	3.995	5.22
**Patient 11**	10y3m	Small mixed breed	8	4	Hypothyroidism	5.591	/	3.48
**Patient 12**	10y3m	Medium mixed breed	16.2	5	Hypercortisolism	3.241	/	4.33
**Patient 13**	13y6m	Labrador retriever	29.4	5	/	6.936	7.263	0.03
**Patient 14**	10y6m	Small mixed breed	6.3	4	Hypercortisolism	2.248	2.015	2.71

**Table 2 animals-11-00408-t002:** Ultrasonographic characteristics of the skin of each dog at T0 and T1. * Dogs that developed macroscopic erythema at T1. (Nothing of relevance = consistent with normal features reported in previous studies.).

	Day 1 of Monitoring (T0)	End of Monitoring (T1)
**Patient 1 ***	Entry echo not well defined and irregular	Entry echo not well defined and irregular
**Patient 2**	Presence of oblique and hyperechogenic bands in the dermis	Presence of oblique and hyperechogenic bands in the dermis
**Patient 3**	Double dermal layer	Entry echo not well defined and irregular, double dermal layer
**Patient 4**	Entry echo not well defined and irregular and double dermal layer	Entry echo not well defined and irregular and double dermal layer
**Patient 5**	Entry echo not well defined and irregular and double dermal layer	Double dermal layer
**Patient 6**	Nothing of relevance	Entry echo not well defined and irregular
**Patient 7**	Entry echo not well defined and irregular and double dermal layer	No ultrasound examination
**Patient 8 ***	Nothing of relevance	Nothing of relevance
**Patient 9**	Presence of oblique and hyperechogenic bands in the dermis and double dermal layer	Entry echo not well defined and irregular, presence of oblique and hyperechogenic bands in the dermis
**Patient 10 ***	Nothing of relevance	Entry echo not well defined and irregular
**Patient 11**	Double dermal layer	No ultrasound examination
**Patient 12**	Nothing of relevance	No ultrasound examination
**Patient 13**	Entry echo not well defined and irregular	Entry echo not well defined and irregular and double dermal layer
**Patient 14**	Double dermal layer	Double dermal layer

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
