# Peer review of "The Influence of Skin Thickness on Flash Glucose Monitoring System Accuracy in Dogs with Diabetes Mellitus"

_animals, 2021, doi:10.3390/ani11020408_

Round 1

Reviewer 1 Report

The authors have addressed my original concerns.

Author Response

Dear Reviewers,

We would like to take the opportunity to thank you for revising this manuscript. The comments you have offered have been helpful for an improvement of the manuscript.

In line with your suggestions, we have revised our manuscript and its new version is enclosed. Point-by-point responses to the comments are listed below.

The changes have been made in the text using bold red font.

REVIEWER 2

Comments and Suggestions for Authors

Introduction

  1. Line 44 Remove the words “small and” and replace with dimensions of sensor in cm or mm.

Done..

Methods

  1. Line 89. Provide more details on the reference method. That is, provide whether it was measured in serum or plasma, the name and manufacturer of the device used to measure it, whether it was from the in-house laboratory or external laboratory, whether there is a published reference, and provide standard deviations and number of measurements used to calculate this bias.

The authors apologize for the oversight. We did not include the reference relative to the following part of the text: “This PBGM underestimates the real BG concentration in the euglycemic and hyperglycemic range with a mean difference between capillary BG measured by the PBGM and serum BG obtained using the reference method (hexokinase method) of -0.26 mmol/l (4.8 mg/dl) and -4.11 mmol/l (74 mg/dL), respectively. Conversely, in the hypoglycemic range, this PBGM overestimates the BG concentration with a mean difference of 0.37 mmol/l (6.8 mg/dl)”. These data are from the same study mentioned in the previous sentence (Fracassi et al.; Ref 15). We have now introduced the reference and part of the information you required (BG measured on serum). The standard deviation has not been provided because it has not been reported in the ref 15. We do not have inserted the other “technical” specifications (name of the manufacturer, external or internal lab, number of measurements used to calculate bias) because we think is enough to provide the reference. However, if you think is more appropriate to insert those data, we can do it.

  1. Line 87“Underestimate” should be underestimates (singular) both times it appears in the sentence.

Done.

  1. Line 117 – it is still not clear, how far the transducer was from the placement of the sensor. More detail needs to be provided as it is not clearly stated. The manuscript states that “the sensor was placed in a clipped and cleaned area on the dorsal side of the neck” and “The transducer was placed halfway between the margin of the occipital bone and the dorsal margin of the scapula.”

Both the sentences concerning the position of the probe and the sensor have been rewritten in order to better clarify this issue (Lines 93-96).

  1. Line 147. Reference this statement “Both of the following minimum criteria for acceptable system accuracy should be met: 1) 95% of the results must be within ±0.8 mmol/L (15 mg/dL) of the BG concentration for a BG concentration <5.5 mmol/L (100 mg/dL) and within ±15% of the BG concentration for a BG concentration >5.5 mmol/L (100 mg/dL) and 2) 99% of the individual BG measured values should fall within zones A and B of the Parkes consensus error grid analysis (EGA) for type 1 diabetes mellitus (DM).

Done.

Results

  1. Line 200 Add text in bold “Considering all the dogs, the mean bias between the glucose concentrations obtained using the FGMS versus those obtained using the PBGM was 3.4 mmol/L (± 2.11) (61 mg/dL±38), and the biases were significantly inversely correlated with the skin thickness (r=-0.6; P=0.02) (Figure 2).

Done.

  1. Line 206 Add text in bold “Figure 2: Correlation between skin thickness and absolute mean bias between the glucose concentrations obtained using the FGMS versus those obtained using the PBGM of each dog.

Done

  1. Line 208 Add text in bold “The mean bias of the FGMS versus the PBGM was in Group 1 was 4.1 (± 2.6) (74 mg/dL ±47) and, in Group 2, it was 0.33 mmol/L (±3.77) (6 mg/dL ±68). The difference was significant (P<0.0001).

Done

 Discussion

  1. Line 278 Change “do” to “due” Indeed, do to technical reasons (i.e. reverberation artifact created by the metallic filament and limited skin surface) it was not possible to perform an US after the application of the sensor

The authors apologize for the oversight. We made the change.

  1. Line 207 Please correct this sentence. Not only is this sentence hard to understand, but the authors confuse the enzymatic method of measuring glucose with bias that occurs associated with different ratios of glucose in erythrocytes and plasma between species ”Therefore, using as a reference the hexokinase method, a lower overestimation and higher underestimation of the IG values would have been observed.

The authors had some difficulties in understanding what the reviewer means with the sentence: “the authors confuse the enzymatic method of measuring glucose with bias that occurs associated with different ratios of glucose in erythrocytes and plasma between species.”

The sentence “Therefore, using as a reference the hexokinase method, a lower overestimation and higher underestimation of the IG values would have been observed” has been modified as follows:

Lines 298-300: “Therefore, in this study, the use of PBGM as reference method might have resulted in a greater overestimation and lower underestimation of the IG values.”  

REVIEWER 3

Comments and Suggestions for Authors

Useful study, considering the increasing popularity of FGMSs. The skin thickness results were very interesting, and I agree that it would have been nice to be able to pinpoint exactly where the sensors were in the skin, but there are obviously technological limitations with ultrasound.

Specific comment:

Line 95: I looked up "trichotomized," and while I think you meant "clipped," I could not find any definition of trichotomized that matched up to this.

The authors thank the reviewer for the suggestion. Trichotomized has been replaced by clipped. 

278: "due" instead of "do"

Done.

Reviewer 2 Report

Introduction

  1. Line 44 Remove the words “small and” and replace with dimensions of sensor in cm or mm.

Methods

  1. Line 89. Provide more details on the reference method. That is, provide whether it was measured in serum or plasma, the name and manufacturer of the device used to measure it, whether it was from the in-house laboratory or external laboratory, whether there is a published reference, and provide standard deviations and number of measurements used to calculate this bias.
  2. Line 87“Underestimate” should be underestimates (singular) both times it appears in the sentence.
  3. Line 117 – it is still not clear, how far the transducer was from the placement of the sensor. More detail needs to be provided as it is not clearly stated. The manuscript states that “the sensor was placed in a clipped and cleaned area on the dorsal side of the neck” and “The transducer was placed halfway between the margin of the occipital bone and the dorsal margin of the scapula.”
  4. Line 147. Reference this statement “Both of the following minimum criteria for acceptable system accuracy should be met: 1) 95% of the results must be within ±0.8 mmol/L (15 mg/dL) of the BG concentration for a BG concentration <5.5 mmol/L (100 mg/dL) and within ±15% of the BG concentration for a BG concentration >5.5 mmol/L (100 mg/dL) and 2) 99% of the individual BG measured values should fall within zones A and B of the Parkes consensus error grid analysis (EGA) for type 1 diabetes mellitus (DM).

Results

  1. Line 200 Add text in bold “Considering all the dogs, the mean bias between the glucose concentrations obtained using the FGMS versus those obtained using the PBGM was 3.4 mmol/L (± 2.11) (61 mg/dL±38), and the biases were significantly inversely correlated with the skin thickness (r=-0.6; P=0.02) (Figure 2).
  2. Line 206 Add text in bold “Figure 2: Correlation between skin thickness and absolute mean bias between the glucose concentrations obtained using the FGMS versus those obtained using the PBGM of each dog.
  3. Line 208 Add text in bold “The mean bias of the FGMS versus the PBGM was in Group 1 was 4.1 (± 2.6) (74 mg/dL ±47) and, in Group 2, it was 0.33 mmol/L (±3.77) (6 mg/dL ±68). The difference was significant (P<0.0001).

 Discussion

  1. Line 278 Change “do” to “due” Indeed, do to technical reasons (i.e. reverberation artifact created by the metallic filament and limited skin surface) it was not possible to perform an US after the application of the sensor
  2. Line 207 Please correct this sentence. Not only is this sentence hard to understand, but the authors confuse the enzymatic method of measuring glucose with bias that occurs associated with different ratios of glucose in erythrocytes and plasma between species ”Therefore, using as a reference the hexokinase method, a lower overestimation and higher underestimation of the IG values would have been observed. “

Author Response

(The authors gave the same response as above.)

Reviewer 3 Report

Useful study, considering the increasing popularity of FGMSs. The skin thickness results were very interesting, and I agree that it would have been nice to be able to pinpoint exactly where the sensors were in the skin, but there are obviously technological limitations with ultrasound.

Specific comment:

Line 95: I looked up "trichotomized," and while I think you meant "clipped," I could not find any definition of trichotomized that matched up to this.

278: "due" instead of "do"

Author Response

Dear Reviewers,

We would like to take the opportunity to thank you for revising this manuscript. The comments you have offered have been helpful for an improvement of the manuscript.

In line with your suggestions, we have revised our manuscript and its new version is enclosed. Point-by-point responses to the comments are listed below.

The changes have been made in the text using bold red font.

REVIEWER 2

Comments and Suggestions for Authors

Introduction

  1. Line 44 Remove the words “small and” and replace with dimensions of sensor in cm or mm.

Done..

Methods

  1. Line 89. Provide more details on the reference method. That is, provide whether it was measured in serum or plasma, the name and manufacturer of the device used to measure it, whether it was from the in-house laboratory or external laboratory, whether there is a published reference, and provide standard deviations and number of measurements used to calculate this bias.

The authors apologize for the oversight. We did not include the reference relative to the following part of the text: “This PBGM underestimates the real BG concentration in the euglycemic and hyperglycemic range with a mean difference between capillary BG measured by the PBGM and serum BG obtained using the reference method (hexokinase method) of -0.26 mmol/l (4.8 mg/dl) and -4.11 mmol/l (74 mg/dL), respectively. Conversely, in the hypoglycemic range, this PBGM overestimates the BG concentration with a mean difference of 0.37 mmol/l (6.8 mg/dl)”. These data are from the same study mentioned in the previous sentence (Fracassi et al.; Ref 15). We have now introduced the reference and part of the information you required (BG measured on serum). The standard deviation has not been provided because it has not been reported in the ref 15. We do not have inserted the other “technical” specifications (name of the manufacturer, external or internal lab, number of measurements used to calculate bias) because we think is enough to provide the reference. However, if you think is more appropriate to insert those data, we can do it.

  1. Line 87“Underestimate” should be underestimates (singular) both times it appears in the sentence.

Done.

  1. Line 117 – it is still not clear, how far the transducer was from the placement of the sensor. More detail needs to be provided as it is not clearly stated. The manuscript states that “the sensor was placed in a clipped and cleaned area on the dorsal side of the neck” and “The transducer was placed halfway between the margin of the occipital bone and the dorsal margin of the scapula.”

Both the sentences concerning the position of the probe and the sensor have been rewritten in order to better clarify this issue (Lines 93-96).

  1. Line 147. Reference this statement “Both of the following minimum criteria for acceptable system accuracy should be met: 1) 95% of the results must be within ±0.8 mmol/L (15 mg/dL) of the BG concentration for a BG concentration <5.5 mmol/L (100 mg/dL) and within ±15% of the BG concentration for a BG concentration >5.5 mmol/L (100 mg/dL) and 2) 99% of the individual BG measured values should fall within zones A and B of the Parkes consensus error grid analysis (EGA) for type 1 diabetes mellitus (DM).

Done.

Results

  1. Line 200 Add text in bold “Considering all the dogs, the mean bias between the glucose concentrations obtained using the FGMS versus those obtained using the PBGM was 3.4 mmol/L (± 2.11) (61 mg/dL±38), and the biases were significantly inversely correlated with the skin thickness (r=-0.6; P=0.02) (Figure 2).

Done.

  1. Line 206 Add text in bold “Figure 2: Correlation between skin thickness and absolute mean bias between the glucose concentrations obtained using the FGMS versus those obtained using the PBGM of each dog.

Done

  1. Line 208 Add text in bold “The mean bias of the FGMS versus the PBGM was in Group 1 was 4.1 (± 2.6) (74 mg/dL ±47) and, in Group 2, it was 0.33 mmol/L (±3.77) (6 mg/dL ±68). The difference was significant (P<0.0001).

Done

 Discussion

  1. Line 278 Change “do” to “due” Indeed, do to technical reasons (i.e. reverberation artifact created by the metallic filament and limited skin surface) it was not possible to perform an US after the application of the sensor

The authors apologize for the oversight. We made the change.

  1. Line 207 Please correct this sentence. Not only is this sentence hard to understand, but the authors confuse the enzymatic method of measuring glucose with bias that occurs associated with different ratios of glucose in erythrocytes and plasma between species ”Therefore, using as a reference the hexokinase method, a lower overestimation and higher underestimation of the IG values would have been observed.

The authors had some difficulties in understanding what the reviewer means with the sentence: “the authors confuse the enzymatic method of measuring glucose with bias that occurs associated with different ratios of glucose in erythrocytes and plasma between species.”

The sentence “Therefore, using as a reference the hexokinase method, a lower overestimation and higher underestimation of the IG values would have been observed” has been modified as follows:

Lines 298-300: “Therefore, in this study, the use of PBGM as reference method might have resulted in a greater overestimation and lower underestimation of the IG values.”  

REVIEWER 3

Comments and Suggestions for Authors

Useful study, considering the increasing popularity of FGMSs. The skin thickness results were very interesting, and I agree that it would have been nice to be able to pinpoint exactly where the sensors were in the skin, but there are obviously technological limitations with ultrasound.

Specific comment:

Line 95: I looked up "trichotomized," and while I think you meant "clipped," I could not find any definition of trichotomized that matched up to this.

The authors thank the reviewer for the suggestion. Trichotomized has been replaced by clipped. 

278: "due" instead of "do"

Done.

This manuscript is a resubmission of an earlier submission. The following is a list of the peer review reports and author responses from that submission.

Round 1

Reviewer 1 Report

General comments

The authors suggest that a sensor that was designed to be placed into the subcutaneous tissue performs better when placed into the dermis. The authors never actually showed that the sensor was indeed placed in the dermis (and actually should not even have needed an US examination to know if the sensor was placed intradermally or in the SC space) but base their conclusion on US examinations, which were not done after the sensors had been placed. Skin thickness in this report in 8 of 14 dogs was >5 mm (at least at one of the measurements), only 1 had a value < 3 mm, whereas literature values (see references below), including a study by one of the authors of this study, show usually values of <2 mm by US or by histological examination. Because the authors must have been aware of the discrepancy between their numbers and the numbers that have been reported in the literature, it is difficult to understand why no effort was made to actually show that the sensor was located in the dermis, and it is unjustifiable to now recommend based on this study that veterinarians place the sensors into the dermis and not into the subcutaneous space contrary to  recommendations by the manufacturer and results from other publications.

Additionally, in this manuscript, conclusions are drawn based on a small number of dogs and a small number of data points, and it is often not even clear which data points were used for some of the calculations. For example, it is not clear what T1 is. It is obvious that T1 means different time points for individual dogs, but it is not clear what time point exactly were used for the calculation and what justification there is to lump all of the data points together. It is not clear what data points were used for the paired t-test. It is also not clear why the authors state that 8 dogs had skin thickness values >5 mm, when in fact only 5 dogs had those values for both measurements (initial and final) and 2 dogs each had one value >5 mm and 1 value <5 mm. So, should the reader assume that in those 2 dogs the sensor had moved out of the dermis into the subcutaneous space during the monitoring? And finally, only initial skin values were recorded in 3 dogs. It is therefore unknown where the sensor might have been during the time of monitoring, intradermally or in the subcutaneous space (if skin thickness values would also be different for the 2 time points in these 3 dogs).

References for skin thickness in dogs.: Diana et al, AJVR 2004; Reme & Dufour, Intern J Appl Res Vet Med 2010;  Heo et al, J Vet Sci 2018; Koenig et al., AJVR, 2016; Mantis et al, Vet. Dermatology, 2014)

Other comments

Line 58: Reference #1 did not examine the effect of BCS on glucose levels, and in reference #2, the number of dogs was much too small to allow such an evaluation.

Line 148-154/160-165: This confusing. It might be helpful to provide skin thickness data, the lifespan of the sensor, and removal date in a table. This would allow the reader to know what the authors considered to be T1 for each dog.

Line 152: Please delete the median wearing time. It is irrelevant because of the large variation.

Line 166: when was the sensor removed in these 3 dogs?

Line 248: This is not correct since 2 of the dogs had values < 5mm

Author Response

REVIEWER 1

General comments to the Author

The authors suggest that a sensor that was designed to be placed into the subcutaneous tissue performs better when placed into the dermis. The authors never actually showed that the sensor was indeed placed in the dermis (and actually should not even have needed an US examination to know if the sensor was placed intradermally or in the SC space) but base their conclusion on US examinations, which were not done after the sensors had been placed. Skin thickness in this report in 8 of 14 dogs was >5 mm (at least at one of the measurements), only 1 had a value < 3 mm, whereas literature values (see references below), including a study by one of the authors of this study, show usually values of <2 mm by US or by histological examination. Because the authors must have been aware of the discrepancy between their numbers and the numbers that have been reported in the literature, it is difficult to understand why no effort was made to actually show that the sensor was located in the dermis, and it is unjustifiable to now recommend based on this study that veterinarians place the sensors into the dermis and not into the subcutaneous space contrary to  recommendations by the manufacturer and results from other publications.

Additionally, in this manuscript, conclusions are drawn based on a small number of dogs and a small number of data points, and it is often not even clear which data points were used for some of the calculations. For example, it is not clear what T1 is. It is obvious that T1 means different time points for individual dogs, but it is not clear what time point exactly were used for the calculation and what justification there is to lump all of the data points together. It is not clear what data points were used for the paired t-test. It is also not clear why the authors state that 8 dogs had skin thickness values >5 mm, when in fact only 5 dogs had those values for both measurements (initial and final) and 2 dogs each had one value >5 mm and 1 value <5 mm. So, should the reader assume that in those 2 dogs the sensor had moved out of the dermis into the subcutaneous space during the monitoring? And finally, only initial skin values were recorded in 3 dogs. It is therefore unknown where the sensor might have been during the time of monitoring, intradermally or in the subcutaneous space (if skin thickness values would also be different for the 2 time points in these 3 dogs).

References for skin thickness in dogs.: Diana et al, AJVR 2004; Reme & Dufour, Intern J Appl Res Vet Med 2010;  Heo et al, J Vet Sci 2018; Koenig et al., AJVR, 2016; Mantis et al, Vet. Dermatology, 2014)

Response

The authors thank the reviewer for the assessment of the manuscript. The Flash glucose monitoring system (FGMS) has already been demonstrated to be accurate enough for use in stable diabetic dogs and dogs with diabetic ketoacidosis. In this study, we aimed to investigate if skin thickness, as measured by US, could affect FGMS accuracy. On the basis of our results, considering the length of the sensor (5mm), we can hypotize that the sensor in dogs with skin thickness> 5mm (in which it has resulted more accurate) probably detect IG in the dermis and not in the subcutaneous tissue (SC) as it was designed for. Unfortunately, we were not able to confirm the exact location of the sensor in the dermis for these reasons: the small filament of the sensor is composed by a metal alloy and therefore it created a riverberation artifact; performing the US after the sensor application would have technically  been difficult due to not enough space. We are aware that FGMS is developed to be placed in the SC tissue and we are not recommending placing the sensor in the dermis (also because the sensor placement occurs automatically by the applicator furnished by the manufacturer). We have just supposed, based on our results, that the sensor in dogs with thicker skin probably reaches the dermis.

In order to better clarify this aspect, the discussion has been modified as follows:

Lines 277-280: “However, this is only speculation because the location of the sensor in the dermis was not confirmed. Indeed, do to technical reasons (i.e.  reverberation artifact created by the metallic filament and limited skin surface) it was not possible to perform an US after the application of the sensor.”

We thank the reviewer for the references below cited. We are aware that our values are higher compared to those reported in the other veterinary studies; however, the majority of these studies have evaluated the skin thickness in healthy/research dogs or in dogs with hyperadrenocorticism (Heo et al, J Vet Sci 2018)  which is an endocrine disease associated with a reduced thickening of the skin. Our study include a population of dogs with naturally occurring diabetes mellitus (DM), therefore a direct comparison with the value reported in the veterinary literature is quite difficult to perform. Moreover, DM, in humans, has been associated with the development of scleredema diabeticorum, characterized by thickening of the skin affecting the posterior neck. Although these dermatologic complications have never been documented in diabetic dogs, similar skin changes might affect diabetic dogs as well and therefore might have been the reason for the higher skin thickness reported in this population of diabetic dogs. In order to clarify this aspect, the following discussion in the text has been added:

Lines 266-273: “Further, the previous studies in which skin thickness in dogs was investigated, were performed in healthy dogs or in dogs with hyperadrenocorticism, an endocrinopathy associated with a reduced skin thickness [28]. In contrast, our study includes dogs with naturally occurring DM, which, in humans is associated with scleredema diabeticorum [29], a dermatologic complication characterized by thickening of the skin affecting the posterior neck. Although this dermatologic complication has never been documented in diabetic dogs, similar skin changes might affect diabetic dogs too and, therefore, might have been the reason of the higher skin thickness reported in this population of diabetic dogs.”

We were aware that the study included a small number of dogs. This aspect, unfortunately, is a limitation of the majority of clinical studies on DM in veterinary medicine. When the comparison between two populations is not significant, this could be related to the low number of cases included (type 2 statistical error). However, because the results are significant, the number of cases included, and the number of paired glucose values obtained, should be considered adequate to draw statistical conclusions.

Concerning the time-point T1, we have introduced a new table (as a supplemental file – also reported in the next paragraph) to help the reader to clarify this aspect. Considering the clinical design of the study, it was difficult to guarantee the same time-point for all dogs (due to different sensor lifespan and owner compliances). Moreover, to classify the dogs in 2 groups, we have only used the values of the skin thickness at T0 in order to avoid the influence of skin inflammation induced by the sensor. This information has been added in the text as follows:

Lines 77-78: “Based on skin thickness measured by ultrasonography (US) at T0 and according to the length of the sensor filament (5mm), the dogs were divided into two groups: dogs with a skin thickness <5 mm (Group 1) and dogs with a skin thickness >5 mm (Group 2).”

Other Comments

  • Line 58: Reference #1 did not examine the effect of BCS on glucose levels, and in reference #2, the number of dogs was much too small to allow such an evaluation.

Response: The sentence “it is not 57 affected by the body condition score (BCS)” has been removed from the text.

  • Line 148-154/160-165: This confusing. It might be helpful to provide skin thickness data, the lifespan of the sensor, and removal date in a table. This would allow the reader to know what the authors considered to be T1 for each dog.

Response: a new table including in the supplementary material containing the lifespan of the sensor, the date of removal and the date of the US examination at T1 was performed has been included. Moreover, we have removed in line 115 the word “immediately” from the following sentence:

“Ultrasonography of the dorsal skin of the neck was carried out by the same experienced sonographer (AD) using a real-time ultrasound machine (Epiq 5G Ultrasound System; Philips Healthcare) equipped with a broadband high-frequency linear-array transducer (5-18 MHz) immediately before the sensor application (T0) and immediately after its removal (T1) in all dogs.”

Table S1: Skin thickness, lifespan of the sensor, removal day of the sensor and day of US examination at T1. NA= not applicable

Skin thickness T0

(mm)

Skin thickness T1

(mm)

Sensor Lifespan (days)

Removal day

(days from sensor insertion)

US of the neck T1 (day)

Patient 1*

3.045

3.073

6

7

7

Patient 2

3.376

3.328

12

14

14

Patient 3

6.303

5.286

10

10

14

Patient 4

5.583

5.255

14

14

14

Patient 5

5.510

4.950

14

14

14

Patient 6

5.858

7.770

14

14

14

Patient 7

6.028

NA

2

NA

NA

Patient 8*

4.675

5.306

13

14

14

Patient 9

6.201

7.023

14

14

14

Patient 10*

4.978

3.995

14

14

14

Patient 11

5.591

NA

10

NA

NA

Patient 12

3.241

NA

10

NA

NA

Patient 13

6.936

7.263

13

14

14

Patient 14

2.248

2.015

10

10

14

  • Line 152: Please delete the median wearing time. It is irrelevant because of the large variation. Response: The median wearing time has been removed.

  • Line 166: when was the sensor removed in these 3 dogs? Response: This information has been underlined in the new table mentioned above (see also comment line 148-154/160-165).

  • Line 248: This is not correct since 2 of the dogs had values < 5mm

Response: This is the reason why we have included the standard deviation. If you prefer, we can report also the range.

Reviewer 2 Report

Major comments

The authors have used a model of naturally occurring cases of diabetes mellitus to answer a clinically relevant question.

I was particularly impressed by the ultrasound based skin thickness determination, which was performed by a published expert.

In addition, the statistical methods are appropriate and the tests that have been employed have been specifically tailored to the different variables.

The results are appropriately contextualised and limitations are highlighted in the discussion.

Minor comments

Table 2: Change "nothing of relevant" to "nothing of relevance"

Change "Not" ultrasound to "No" ultrasound

Line 279 onwards: The authors assert that thin skin causes an overestimation of the IG, but then proceed to explain the reason by citing what happens in dogs with thick skin. They are welcome to state what happens in thick skinned dogs, but not as a reason why thin-skinned dogs cause overestimation. Consequently, the flow of the argument is jarring and should be re-ordered.

Line 302: These are new results that suddenly appear in the discussion for the first time. I suggest that they are first mentioned in the results section and then discussed in the discussion. 

Line 315: In this study, no dog was anemic; i.e replace none with no.

Line 340. I would recommend ending the last sentence with...whenever unexpected results are obtained, especially in dogs with skin thickness < 5 mm. 

Author Response

REVIEWER 2

Major Comments

The authors have used a model of naturally occurring cases of diabetes mellitus to answer a clinically relevant question. I was particularly impressed by the ultrasound based skin thickness determination, which was performed by a published expert. In addition, the statistical methods are appropriate and the tests that have been employed have been specifically tailored to the different variables.

The results are appropriately contextualised and limitations are highlighted in the discussion.

Response: Authors thank the reviewer for the positive assessment of the manuscript.

Minor comments

  • Table 2: Change "nothing of relevant" to "nothing of relevance" Response: Thank you for the suggestion. “Relevant” has been change to “relevance”.

  • Change "Not" ultrasound to "No" ultrasound

Response: Done.

  • Line 279 onwards: The authors assert that thin skin causes an overestimation of the IG, but then proceed to explain the reason by citing what happens in dogs with thick skin. They are welcome to state what happens in thick skinned dogs, but not as a reason why thin-skinned dogs cause overestimation. Consequently, the flow of the argument is jarring and should be re-ordered.

Response: The authors thank the reviewer for the right observation. The sentence “In particular, in dogs with thinner skin, an overestimation of the IG measurements detected by the FGMS was observed” has been removed from the text.

  • Line 302: These are new results that suddenly appear in the discussion for the first time. I suggest that they are first mentioned in the results section and then discussed in the discussion. 

Response: In lines 174-175 (results section) it is reported that 3 dogs developed erythema in the text: “At T1, 3/14 dogs showed mild self-limiting erythema at the point of the sensor application.”

  • Line 315: In this study, no dog was anemic; i.e replace none with no.

Response: Thank you for the suggestion. “none” has been replaced with “no”.

  • Line 340. I would recommend ending the last sentence with...whenever unexpected results are obtained, especially in dogs with skin thickness < 5 mm. 

Response: Thank you for the suggestion. The problem is that in clinical practice is not possible to measure the skin thickness routinely. Therefore, given the sentence preceding that in question: “Considering the difficulty in predicting skin thickness on the basis of dog size”, if you agree, we prefer to not modify the text.

Reviewer 3 Report

This paper evaluated if the skin thickness where the sensor was applied can influence the FGMS Freestyle precision in diabetic dogs. Although we have excellent articles so far evaluating this system in dogs, none of them assessed the skin thickness, and the present study showed that it influences the results. This way, I consider that studies like this have great importance, evaluating other variables that may influence the precision of this FGMS that shows itself as promissory in veterinary medicine.

I consider the absence of data in the hypoglycemic range as the main limitation of the present study

As stated in the specific comments, there are some important details that need to be corrected.

Specific comments:

Lines 64 - 66 – The hydration can influence the skin thickness and the precision of FGMS. This way, it is relevant to describe better the animals in this experiment, if they presented meaningful changes to physical tests and/or laboratory at the beginning and during this study since that the hydration was not an evaluated variable.

Lines 90 - 91 – Describe in detail how the sensor was attached to the animal’s body, saying which adhesive tape was used for that since it is relevant information to use in future papers. The reader cannot be asked to retrieve ref. Corradini et al. (2016) to know how this was done.

Line 151 – Explain what “ERR” is.

Lines 191 - 194 – What is the number of data samples paired obtained on the 1st, 7th, and 14th days? Although it is not the main objective of this paper, it would be meaningful to evaluate if there was a difference in the FGMS precision among the days.

Line 339 – I suggest you to add the "BCS", since this variable did not show itself reliable to predict skin thickness either:

“Considering the difficulty in predicting skin thickness on the basis of dog size and BCS, the Authors recommend...”

Figure 2 – I suggest you to add a trendline to the graph.

Author Response

REVIEWER 3

Comments and Suggestions for Authors

This paper evaluated if the skin thickness where the sensor was applied can influence the FGMS Freestyle precision in diabetic dogs. Although we have excellent articles so far evaluating this system in dogs, none of them assessed the skin thickness, and the present study showed that it influences the results. This way, I consider that studies like this have great importance, evaluating other variables that may influence the precision of this FGMS that shows itself as promissory in veterinary medicine.

I consider the absence of data in the hypoglycemic range as the main limitation of the present study

As stated in the specific comments, there are some important details that need to be corrected.

Response: We fully agree with the reviewer that one of the main limitations of this study is the low number of samples in the hypoglycemic range (this, unfortunately, is a limit of the majority of similar clinical study).

Specific comments:

  • Lines 64 - 66 – The hydration can influence the skin thickness and the precision of FGMS. This way, it is relevant to describe better the animals in this experiment, if they presented meaningful changes to physical tests and/or laboratory at the beginning and during this study since that the hydration was not an evaluated variable.

Response: We fully agree with the reviewer’s comment. In a recent study, FGMS accuracy resulted affected by the hydration status (lower accuracy in dehydrated animals) in unstable diabetic dogs (Evaluation of a flash glucose monitoring system in dogs with diabetic ketoacidosis. Silva et al. 2020). When we have designed our study, we didn’t consider this aspect and ,unfortunately, the hydration status of the dogs was not objectively registered. Based on our medical records, all dogs included in this study were normohydrated, however, considering that the evaluation of the hydration state was not performed according to a standardized scheme, if you agree, we think is more appropriate not to introduce this aspect in the text. However, this issue can be introduced in the limitations of the study as follows: Lines 355-357: “Further, the hydration state can affect skin thickness and FGMS accuracy [50], however, in this study the hydration state of the dogs has not been assessed, preventing additional analysis.”

  • Lines 90 - 91 – Describe in detail how the sensor was attached to the animal’s body, saying which adhesive tape was used for that since it is relevant information to use in future papers. The reader cannot be asked to retrieve ref. Corradini et al. (2016) to know how this was done.

Response: A more detailed description of the sensor application has been added to the text in the material and methods section:                                                                                                         

Lines 95-99:”In particular, an area of around 5 x 5 cm was trichotomized, cleaned with chlorhexidine and 70% alcohol. The sensor was inserted with the aid of the applicator provided by the manufacturer. After positioning, it was fixed with extra tape (PIC Solution Soft Fix, Pikdare S.p.A., Casnate con Bernate, Como, Italy), and a cotton and an elastic body bandage (Vetrap RM, 3M Italia Srl, Pioltello, Milano, Italy) were used to secure the sensor to the body [1].”

  • Line 151 – Explain what “ERR” is.

Response: The meaning of ERR has been added in the text as follow:  

Lines 155: “ERR” (the reader is unable to communicate with the sensor).

  • Lines 191 - 194 – What is the number of data samples paired obtained on the 1st, 7th, and 14th days? Although it is not the main objective of this paper, it would be meaningful to evaluate if there was a difference in the FGMS precision among the days.

Response: Thank you for the suggestion. The main aim of this study was to evaluate if skin thickness can affect FGMS accuracy. The accuracy over time has already been evaluated in the other two studies (Corradini et al., 2016 and Silva et al. 2020). Considering the small sample size (T1=49 paired values, T7=52 paired values, T14=26 paired values), we think that adding this new statistical analysis would lead to limited information. Therefore, if you agree, we prefer to remain faithful to the primary objective and to not introduce this new statistical analysis. However, if you think it is important, we can do also this analysis.

  • Line 339 – I suggest you to add the "BCS", since this variable did not show itself reliable to predict skin thickness either:

Response: The BCS has been added.                  

Lines 366-367: “Considering the difficulty in predicting skin thickness on the basis of dog size and BCS, the Authors recommend checking BG concentration whenever unexpected FGMS results are obtained.” 

  • Figure 2 – I suggest you to add a trendline to the graph.

Response: Done

Reviewer 4 Report

Introduction

  1. Remove the subjective word ‘comfortable” because this is subjective and you don’t know how comfortable it is for a dog. Also replace “easy and quick” with more “rapid” as it is less subjective in this sentence "It consists of a small and comfortable round sensor, which continuously measures the glucose in the interstitial fluid through a small filament inserted under the skin, allowing easy and quick glucose monitoring."

Methods

  1. The Optium Xceed is a glucose meter calibrated for human blood. You need to provide data on the bias compared to use of a PBGM calibrated for dog blood or glucose measured by an automated chemistry analyser in a veterinary laboratory. You are using a PBGM which already has a bias because it is not calibrated for dog blood, so your data in terms of bias with the FGMS are somewhat irrelevant. Clinicians need to know the bias with glucose measured by a method calibrated for dog blood..
  2. Change little to small in the sentence: The sensor measures the IG concentration through a little catheter
  3. Add greater or lesser signs to all values mentioned in this sentence on line 95 “1 mmol/L (£20 mg/dL) and ≥27.7 mmol/L (500 mg/dL),”
  4. It is unclear where in relation to the sensor placement that the ultrasound measurements were made. It is also unclear if T0 was made directly before or after the sensor placement and at what distance from the sensor.
  5. What does “nothing relevant” mean in the table?
  6. Is table 2 of value, given it takes a whole page? Is it necessary?

Results

  1. Make it clear to the reader which measurements for what device you are comparing with in this sentence and make it easier for the reader to understand as shown “Considering the ISO 15197:2013 requirements, in Group 1, the percentage of values measured by FBGM that were within ± 0.83 mmol/L (15 mg/dL) of the BG concentration measured with the PBGM for BG concentrations <5.55 mmol/L (100 mg/dL) was 0% (0/4), and the percentage that were within ±15% of the PBGM BG concentration for BG concentration ≥5 mmol/L (100 mg/dL) was 32% (14/44).
  2. Line 191 – 4 – in this sentence be very clear what you were comparing with what to get the bias and whether it was positive or negative difference (more or less) “Considering all the dogs, the mean bias was 3.4 mmol/L (± 2.11) (61 mg/dL±38), and the biases were significantly inversely correlated with the skin thickness (r=-0.6; P=0.02) (Figure 2).
  3. You say what A, B, C etc mean in the legend for the figure but these also need to be described for the reader in the text in this sentence “Evaluation of the data using the Parkes consensus EGA showed that 94% (45/48) and 99% (78/79) of the FGMS results fell in zones A+B in Groups 1 and 2, respectively (Figure 4). These percentages did not differ significantly (P=0.15).

Discussion

  1. In this discussion you need to also discuss the difference between meters calibrated for canine blood and the bias with the meter you used. You need to also provide evidence of bias in dog blood of the difference in methods between hexokinase and glucokinase:”These differences may be due to the fact that, in the study of Corradini et al., the accuracy of the FGMS was evaluated by comparing the glucose measurements of the FGMS with the plasma glucose measured using the hexokinase method whereas, in the present study, as well as in the Malerba et al. study, the same PBGM was used as a reference.”
  2. Discuss how your results might have looked if you had used a PBGM that was calibrated for dog blood and did not have a bias to underestimate blood glucose concentration.
  3. Add to discussion of limitations that PBGM was not calibrated for dogs
  4. Discuss mechanism why the PBGM used underestimates blood glucose in dog’s blood
  5. Add to discussion of limitations on small sample size not just that it reduced power but reduces confidence in the results

Author Response

REVIEWER 4

Introduction

  1. Remove the subjective word ‘comfortable” because this is subjective and you don’t know how comfortable it is for a dog. Also replace “easy and quick” with more “rapid” as it is less subjective in this sentence "It consists of a small and comfortable round sensor, which continuously measures the glucose in the interstitial fluid through a small filament inserted under the skin, allowing easy and quick glucose monitoring."

Response: Thank you for the suggestion. The word comfortable has been removed and “easy and quick” has been replaced by “more rapid” as follows:  Lines 40-42: “It consists of a small and round sensor, which continuously measures the glucose in the interstitial fluid through a small filament inserted under the skin, allowing more rapid glucose monitoring.”

Methods

  1. The Optium Xceed is a glucose meter calibrated for human blood. You need to provide data on the bias compared to use of a PBGM calibrated for dog blood or glucose measured by an automated chemistry analyser in a veterinary laboratory. You are using a PBGM which already has a bias because it is not calibrated for dog blood, so your data in terms of bias with the FGMS are somewhat irrelevant. Clinicians need to know the bias with glucose measured by a method calibrated for dog blood..

Response: The mean difference between PBGM and reference method measured glucose valued has been added in the text as follows:  Lines 87-91: “This PBGM underestimate the real BG concentration in the euglycemic and hyperglycaemic range with a mean difference between capillary BG measured by the PBGM and BG obtained using the reference method of -0.26 mmol/l (4.8 mg/dl) and -4.11 mmol/l (74 mg/dL), respectively. Conversely, in the hypoglycemic range, this PBGM overestimate the BG concentration with a mean difference of 0.37 mmol/l (6.8 mg/dl).” 

  1. Change little to small in the sentence: The sensor measures the IG concentration through a little catheter

Response: Done.

  1. Add greater or lesser signs to all values mentioned in this sentence on line 95 “1 mmol/L (£20 mg/dL) and ≥27.7 mmol/L (500 mg/dL),”

Response: Done.

  1. It is unclear where in relation to the sensor placement that the ultrasound measurements were made. It is also unclear if T0 was made directly before or after the sensor placement and at what distance from the sensor.

Response: The ultrasound (US) examination has been performed immediately before the sensor application (T0). This has been already specified in the text (See line 115).  The US measurements have been performed in the area included halfway between the margin of the occipital bone and the dorsal margin of the scapula. The area of the sensor placement (corresponded to the ultrasonographic food-print of the probe) has been drawn with a marker and, consequently, the sensor has been applied in this specific area.

  1. What does “nothing relevant” mean in the table?

Response: It means that the skin aspect was as that reported in normal dogs. This issue has been added in the caption of the table. (Nothing of relevance=consistent with normal featureas reported in previous studies).

  1. Is table 2 of value, given it takes a whole page? Is it necessary?

Response: we think that the ultrasonographic appearance of the skin in the two different time points could add important information for the readers but, if the reviewer disagree, we can moved the table from the main text to supplementar material.

Results

  1. Make it clear to the reader which measurements for what device you are comparing with in this sentence and make it easier for the reader to understand as shown “Considering the ISO 15197:2013 requirements, in Group 1, the percentage of values measured by FBGM that were within ± 0.83 mmol/L (15 mg/dL) of the BG concentration measured with the PBGM for BG concentrations <5.55 mmol/L (100 mg/dL) was 0% (0/4), and the percentage that were within ±15% of the PBGM BG concentration for BG concentration ≥5 mmol/L (100 mg/dL) was 32% (14/44).

Response: In the materials and methods section (lines 145-147) the values used to do the Error Grid Analysis is specified as follows: “Analytical and clinical accuracy within the 2 groups of dogs was evaluated by comparing the results of the PBGM measurements and those obtained from the FGMS using the ISO 15197:2013 criteria.”

If the reviewer thinks this is not sufficiently clear, we can reformulate the sentence.

  1. Line 191 – 4 – in this sentence be very clear what you were comparing with what to get the bias and whether it was positive or negative difference (more or less) “Considering all the dogs, the mean bias was 3.4 mmol/L (± 2.11) (61 mg/dL±38), and the biases were significantly inversely correlated with the skin thickness (r=-0.6; P=0.02) (Figure 2).

Response: In the materials and methods section (lines 138-142) has been specified how the bias was calculated “The bias, defined as the absolute value of the difference between interstitial glucose concentration obtained with the FGMS and corresponding blood glucose concentrations obtained with the PBGM, was calculated.  For each dog, the mean bias was calculated as the mean of the biases obtained from all the paired values and it was correlated to the mean skin thickness using the Pearson test.” If the reviewer think is not clear, the authors can reformulate the sentence.

  1. You say what A, B, C etc mean in the legend for the figure but these also need to be described for the reader in the text in this sentence “Evaluation of the data using the Parkes consensus EGA showed that 94% (45/48) and 99% (78/79) of the FGMS results fell in zones A+B in Groups 1 and 2, respectively (Figure 4). These percentages did not differ significantly (P=0.15).

Response: Done

Discussion

  1. In this discussion you need to also discuss the difference between meters calibrated for canine blood and the bias with the meter you used. You need to also provide evidence of bias in dog blood of the difference in methods between hexokinase and glucokinase:”These differences may be due to the fact that, in the study of Corradini et al., the accuracy of the FGMS was evaluated by comparing the glucose measurements of the FGMS with the plasma glucose measured using the hexokinase method whereas, in the present study, as well as in the Malerba et al. study, the same PBGM was used as a reference.”
  2. Discuss how your results might have looked if you had used a PBGM that was calibrated for dog blood and did not have a bias to underestimate blood glucose concentration.

Response: The authors thank the reviewer for the right observations and suggestions. However, we think is more appropriate to discuss this aspect comparing the human PBGM used in this study with the hexokinase methods, which is the gold standard to measure glycemia. Therefore, if the reviewer agrees, we discuss this aspect comparing the human PBGM with the hexokinase method as follows: Lines 293-298: “Human PBGM, as the one used in this study, are plasma calibrated to enable better comparison between laboratory and PBGM device measurements. As the amount of glucose within erythrocytes in dogs is lower than in humans, the human PBGM device tend to underestimate the “true” BG concentration. Therefore, using as a reference the hexokinase method, less overestimation and higher underestimation of the IG values would have been observed.”

  1. Add to discussion of limitations that PBGM was not calibrated for dogs

Response: This has been added in the text as follows:

Lines 335-336: “There were several limitations in the present study. First, capillary BG concentrations obtained using a PBGM not calibrated for dogs, and not using the hexokinase method, were used as a reference to evaluate the accuracy of the FGMS”

  1. Discuss mechanism why the PBGM used underestimates blood glucose in dog’s blood.

Response: Thank you for the suggestion. This has been included in the discussion in lines 293-298.

  1. Add to discussion of limitations on small sample size not just that it reduced power but reduces confidence in the results.

Response: Thank you for the suggestion. This has been underlined in the text as follows:

Lines 340-341: “Second, the small number of dogs included in the study could have influenced the power of the statistical analysis and might have affected the reliability of the results.”
